# Discontinuation, persistence and adherence to subcutaneous biologics delivered via a homecare route to Scottish adults with rheumatic diseases: a retrospective study

Samantha Alvarez-Madrazo,[1,2] Kimberley Kavanagh,[3] Stefan Siebert,[4] Yvonne Semple,[5,6] Brian Godman,[7,8] Alessandra Maciel Almeida,[9] Francisco de Assis Acurcio,[9] Marion Bennie[1,10]

For numbered affiliations see end of article.

**Correspondence to**
Professor Marion Bennie;
marion.bennie@strath.ac.uk

## ABSTRACT

**Objectives** To understand patterns of subcutaneous (SC) biologics use over time in adults with inflammatory rheumatic musculoskeletal diseases receiving a homecare delivery service.

**Design** Retrospective cohort.

**Setting** Patients in secondary care receiving SC biologics in the largest Scottish Health Board.

**Participants** A new bespoke cohort was created from routine data gathered as part of a health board Homecare Service Database. Patients over 18 years who received a supply of SC biologic from January 2012 to May 2015 with a diagnosis for rheumatoid arthritis (RA), psoriatic arthritis (PsA) or ankylosing spondylitis (AS) were included.

**Outcomes measured** A standardised framework was applied by measuring discontinuation rates, persistence using Kaplan-Meier analysis and Cox regression and adherence using medication refill adherence (MRA) and compliance rate (CR).

**Results** 751 patients were identified (AS: 105, PsA: 227, RA: 419) of whom 89.3% had more than one biologic delivery (median days' follow-up: AS: 494; PsA: 544; RA: 529) and 83.2% did not switch biologic. For all conditions, approximately half were persistent on their index biologic (52% AS, 54% PsA, 48%RA). Of patients who discontinued treatment, the majority reinitiated with the same biologic (19% AS, 18% PsA and 21% RA). Overall adherence during the period of treatment was over 80% when calculated using MRA (median %MRA: AS: 84.0%, PsA: 85.0%, RA: 82.4%) or CR (median %CR: AS: 96.6%, PsA: 97%, RA: 96.6%).

**Conclusion** Use of linked routine data is a sustainable pathway to enable ongoing evaluation of biologics use. A more consistent approach to studying use (discontinuation, persistence and adherence metrics) should be adopted to enable comparability of studies.

## Strengths and limitations of this study

► This is the first real-world cohort containing type, frequency and dose of seven subcutaneous biologics delivered directly to patients with ankylosing spondylitis, psoriatic arthritis and rheumatoid arthritis.

► The cohort is sustainable being generated as part of routine clinical care, can be easily updated and linked to other administrative datasets and provides a novel way to assess adherence, persistence and discontinuation of biologics.

► The limitations include having a mixed cohort of naive and existing biologic users, not capturing reasons for variability in patients' response, use of intravenous biologics or prior use of any biologic or subcutaneous methotrexate.

► The generalisability of the results in a Scottish setting has to be tested, as this cohort covers the largest Scottish Health Board.

spondylitis (AS) are characterised by inflammatory joint symptoms requiring long-term management. Over the past two decades, the use of biologic therapies has revolutionised their treatment, with national and global groups recommending their use in patients with moderate to severe disease unresponsive to conventional therapies.[1–5] While clinical trials have demonstrated their short-term efficacy and safety, long-term evidence requires observational studies and real-world data. Despite reassuring results to date,[6–10] important challenges and unanswered questions remain.

Furthermore, despite the apparent shared inflammatory mechanisms and therapies, there are significant differences between these inflammatory conditions in terms of prevalence, gender and sex distribution,

## INTRODUCTION

Chronic inflammatory rheumatic musculoskeletal diseases such as rheumatoid arthritis (RA), psoriatic arthritis (PsA) and ankylosing

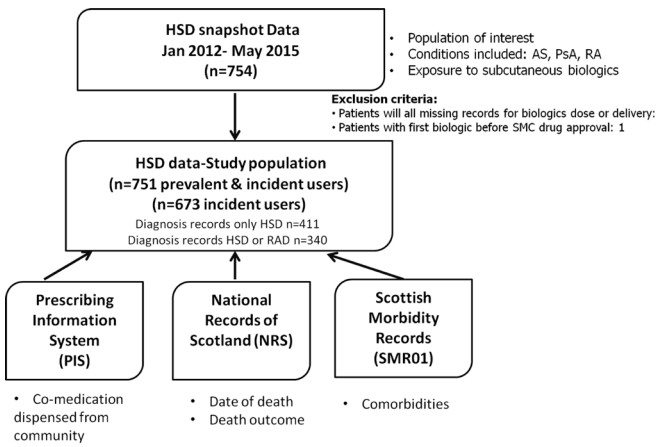

**Figure 1** Variables obtained from different Scottish datasets. AS, ankylosing spondylitis; HSD, Homecare Service Database; PsA, psoriatic arthritis; RA, rheumatoid arthritis; RAD, Rheumatology Arthritis Database; SMC, Scottish Medicines Consortium.

genetics, pathophysiology and clinical phenotype, including extra-articular manifestations and comorbidities.[11–16] Therefore, the safety and long-term therapy data for one condition (usually RA due to the larger numbers and longer pharmacological history) cannot simply be assumed to apply to the other conditions, even for the same drug. It is therefore important to evaluate these factors in real-world settings across the range of conditions for which these drugs are used.

In addition to drug and disease-related factors, in clinical practice, poor adherence and persistence to therapy are important detrimental factors, contributing to treatment failure and disease progression.[17] Most of the direct comparisons for adherence and persistence are from observational studies, mainly focusing on the first biologic to reach the market/clinical practice (infliximab, adalimumab and etanercept).[17–19] A few studies have incorporated other biologics,[20–26] with more biologics set to be approved in the upcoming years. Although the standardised measurement of adherence and persistence, dividing the process in three phases—initiation, implementation and discontinuation—has been advocated by Vrijens *et al* and endorsed by the European Society for Patient Adherence, Compliance and Persistence (ESPACOMP),[27] this methodology has mainly been applied to oral medications and not to biologics, which are administered subcutaneously or intravenously, with complex dosing schedules.

Disease or drug registers are expensive, labour intensive and not optimal for evaluating adherence and persistence, so other methods are required.[17 18] In many countries, administrative or prescription datasets linked to reimbursement have been used,[28–31] but these do not always indicate whether the participant actually received the medication. Furthermore, biologics usually require specialist prescription so are not routinely captured in primary care prescription datasets.

This study aimed to use a homecare service database for subcutaneous biologics to understand the patterns of their use over time in a population of adults with rheumatic diseases.

## METHODS

### Data sources

For this study, a new bespoke, retrospective cohort of patients receiving subcutaneous biologics was created by linking administrative electronic health records at a patient level via a unique national (Scottish) health service identifier, known as the Community Health Index.[32] This dataset included two resources available in the largest Scottish Health Board, NHS Greater Glasgow and Clyde (GGC), with a population covering 1.2 million of the 5.4 million residents in Scotland.[33] The two novel resources were the Homecare Service Database (HSD) and the local Rheumatology Arthritis Database (RAD), in addition to three existing nationally held datasets: the Prescribing Information System (PIS),[34] the mortality records (National Records of Scotland (NRS)) and the Scottish Morbidity Records (SMR01).[35] The variables obtained from each dataset are outlined in figure 1.

Of the GGC patients receiving biologics up until early 2018, 98% received subcutaneous biologics uniquely via the homecare delivery route, with delivery details captured in the HSD and managed through the pharmacy distribution system in the health board. Since 2012, the HSD has systematically captured information about patients (eg, hospital attended, indication for biologics, drugs and dose, administration and delivery frequency, date and quantity delivered), prescribers and healthcare delivery companies. The remaining 2% of patients receiving intravenous biologics are not captured in the HSD as these medicines are administered directly in hospital day wards.

The RAD contains information of patients who have attended rheumatology clinics in two of the four main hospitals in Glasgow since 1997. Diverse clinical variables are recorded, including the main and secondary diagnoses, biologic screening/follow-up, comorbidities, other medications and lifestyle factors.

The PIS includes prescriptions prescribed, dispensed and reimbursed within the community setting,[34] NRS captures cause-specific mortality records and SMR01 has episode-level data on all hospital inpatient and day care episodes from hospitals in Scotland.[35]

### Study population

Patients who received at least one delivery of subcutaneous biologics in the HSD dataset with a diagnosis for AS, PsA or RA were eligible for this study. The RAD database was used to validate the diagnosis in a subset of HSD patients (see online supplementary file and supplementary figure 1 for validation methods and results).

All patients who had a biologic supply date between January 2012 and May 2015 and were at least 18 years

old at cohort entry were included. All patients were followed-up until death, migration to a primary care practice outside Scotland or end date of the study (May 2015), whichever occurred first. In the overall cohort, both prevalent and incident users were included (n=751, 105 AS, 227 PsA and 419 RA). To illustrate the stability of findings, a subset analysis was carried out with only incident users (n=673, 95 AS, 208 PsA and 370 RA), defined as patients who did not receive a subcutaneous biologic in the first 6 months of the study period (from January to June 2012) (online supplementary file).

One patient was excluded as its first biologic delivery was prior to the date of approval for the drug by the Scottish Medicines Consortium, which is the national health technology assessment process for all new medicines in Scotland.[36] Also two patients having all missing records in the HSD for dose, frequency or how often the delivery of the biologics would take place were deleted to enable calculations of weeks covered per delivery in the rest the cohort.

The first delivery date for any biologic was established as the index date and the first biologic recorded was considered the index biologic. Age was calculated at index date. Socioeconomic status was assessed using the Scottish Index of Multiple Deprivation quintile (SIMD) 2012 quintiles, a measure that incorporates different aspects of deprivation into a single index. Patients with missing SIMD values remained in the cohort. To assess comedication, patients with at least one prescription in the 12 months prior to the index date in PIS for antibacterials (British National Formulary (BNF) section 5.1), antifungals (BNF section 5.2), antivirals (BNF section 5.3), oral methotrexate (prescribed item methotrexate), disease-modifying antirheumatic drugs (DMARDs; BNF subsection 10.1.3), non-steroidal anti-inflammatory drugs (NSAIDs; BNF subsection 10.1.1) and corticosteroids (BNF subsections 1.5.2 or 6.3.2) were recorded.

### Data analyses

Patients characteristics were summarised using descriptive statistics. Medians and IQRs are provided for continuous variables, and global comparisons were done using the Kruskal-Wallis test. Numbers and percentages are provided for categorical variables, and global comparisons were done using either $\chi^2$ test or Fisher's exact test (in cases where cells had expected frequencies <5). To protect confidentiality in the results, the GGC Safe Haven carried out a statistical disclosure control. The cell suppression method was applied when the cell values were small and there was a risk of disseminating sensitive information.[37]

In this study, we advocate the use of a standardised framework that includes measurements of discontinuation, persistence and adherence.[27 38]

Considering the variation in the administration frequencies and number of injections delivered according to biologic, first the estimated days covered per delivery was calculated by multiplying the quantity delivered by the standard administration frequency according to the approved licence for each biologic. Knowing the delivery date and the estimated days covered allowed estimation of the expected end date covered and how it related with the following delivery and whether there were gaps or not. These metrics were the basis of the utilisation analysis.

Discontinuation was defined as the end of treatment, either because there was a considerable gap of more than 56 days between deliveries (reinitiation) or no more deliveries were received (cessation). Those who did not discontinue were defined as persistent. Discontinuation rates were estimated using the refill-gap method.[39] Patients were censored at first discontinuation, predefined as a gap of more than 56 days with no biologics delivery after exhaustion of the days' supplied by the previous delivery, regardless of whether patients switched to another biologic. This gap was selected because most biologics were delivered bimonthly, approximately five times the half-life for the biologics of interest would have elapsed, and this was comparable with other studies.[26 29] A sensitivity analysis was conducted using a gap of 28 days (minimum period between deliveries) and 84 days (maximum period between deliveries) (see online supplementary file).

Crude persistence was estimated using Kaplan-Meier analysis, and Cox regression was used to assess persistence postindex date, in patients with at least 6 months follow-up. The proportional hazards (PHs) assumption for a Cox regression model was tested in both the crude and the adjusted model and stratified where appropriate to ensure a non-significant global p value.[40] Patients who died or migrated were censored.

The hazard ratios (HRs) for persistence with biologics were adjusted for sex, age, SIMD, use of index biologic or not (meaning a switch could have happened while being persistent), use of oral methotrexate in the 12 months before to the index date and concomitant use of oral methotrexate or DMARD during study period.

Adherence is defined as to what extent the biologics delivered correspond to the treatment regimen recommended in the BNF. There is no consensus in the literature on how best to estimate adherence thus two measurements were used for patients with at least two deliveries of the index biologic: medication refill adherence (MRA) and compliance rate (CR).[38 41] MRA gives an overall adherence percentage describing the exposure during the whole study period regardless of the time when the last prescription was delivered and is calculated as (total days' supply/total days in study) × 100. In contrast, CR describes the exposure between first and last prescription delivered (total days' supply excluding last delivery/days from first delivery up to but not including last delivery) × 100.

Analyses were performed in R software, V.3.5.0.[42]

### Patient and public involvement

The prelimary design of this study was presented to the Farr Scotland Public Panel. According to their experience and preferences, the research questions and outcome

measures were developed to make the results generalisable to patients with rheumatic conditions.

## RESULTS

The study included 751 patients: 105 with AS, 227 with PsA and 419 with RA (table 1). From them, 673 patients (89.6%) were incident users, 95 AS (90.5%), 208 PsA (91.6%) and 370 RA (88.3%). The patients with RA were older and had a higher proportion of females. Regardless of diagnosis, the majority of patients receiving biologics resided in the most deprived areas.

From the overall cohort, a total of 89.3% of patients had more than one delivery of biologic during the study period and 83.2% did not switch biologic. Patients were most commonly started on adalimumab or etanercept as index biologic. The median days' follow-up on biologic therapy, regardless of whether the patients switched to others biologics, was: 494 days in patients with AS; 544 days in patients with PsA and 529 days in patients with RA.

Use of concomitant non-biologic medication during the study period was more frequent among patients with RA with oral methotrexate (54.9%), any DMARDs (81.6%) and NSAIDs (63.2%) most commonly prescribed. A similar pattern of medication use was observed in patients with PsA. In contrast, in patients with AS, NSAIDs (71.4%) were the most commonly coprescribed medications. Prescription of at least one course of antibiotics in patients during the study period was similar for the three conditions (RA: 62.5%, PsA: 59.9%, AS: 49.5%). Similar trends of concomitant medications were found in the incident cohort (see online supplementary table 1).

### Discontinuation and persistence

For all three conditions, approximately half of the patients were persistent on their index biologic throughout the study period (52% AS, 54% PsA, 48% RA). This was also observed in the incident cohort (55% AS, 55% PsA, 49% RA) as shown in figure 2 for the overall cohort and online supplementary figure 2.

Some patients after a period of discontinuation reinitiated the same biologic received at the index date (overall cohort 19% AS, 18% PsA and 21% RA; incident cohort 18% AS, 18% PsA, 19% RA).

A small proportion of patients switched. There were patients changing their index to another biologic either while being persistent (overall cohort 4% AS, 9% PsA and 5% RA; incident cohort 4% AS, 9% PsA and 6% RA) or reinitiating after a discontinuation, with a gap longer than 56 days between deliveries (overall cohort 7% AS, 4% PsA and 1% RA; incident cohort 5% AS, 3% PsA and 1% RA).

The remaining patients had a discontinuation and ceased treatment, meaning they did not restart a subcutaneous biologic during the study period (overall and incident cohort 18% AS, 15% PsA and 25% RA). A sensitivity analysis was conducted using a gap of 28 and 84 days

showing similar trends (results for overall and incident cohorts are in online supplementary figure 3).

The crude survival curve of *time to discontinuation* (first gap greater than 56 days between biologic deliveries) of any biologic is shown in figure 3. In the crude model, the proportion of patients persisting is slightly higher in PsA and AS compared with RA; however, there is no significant difference according to the log-rank test (p=0.0983). The same results were observed in the incident cohort (p=0.165; online supplementary figure 4). The PH assumption was not violated with all the patients (global p=0.757) or in the incident cohort (global p=0.953).

After adjustment for covariates, there was a slight difference between patients with RA and PsA (HR=0.71, 95% CI 0.54 to 0.95, p=0.02). Lower risk of discontinuation was associated with the two least deprived quintiles (SIMD 4 or 5), whereas female sex and SIMD 3 (deprived quintile) were associated with a higher risk of discontinuation as shown in figure 4. Age, SIMD 2 (more deprived quintile), prior use of of oral methotrexate, concomitant use of oral methotrexate or DMARD were not statistically signficant covariates and did not influence the model, which excluded 21 patients with missing SIMD values. A subanalysis was performed with the incident cohort in which there was no longer a difference between sex and SIMD 4; however, the results of rheumatic conditions and SIMD 3 and SIMD5 quintiles remained significant (see online supplementary figure 5). As those who used the index biologic or switched was used as a stratifying variable, no effect size is estimated for this covariate. The PH assumption was not violated when estimating the adjusted model with all the patients (global p=0.545) or the incident cohort (global p=0.192).

### Adherence

Patients with at least two deliveries of the first (index) biologic (n=671, 89.3%) were used to evaluate adherence, calculated using two methods.

Overall adherence during the period of treatment was over 80% when calculated using either the MRA or CR. However, this was consistently higher using %CR (median (IQR) for AS 96.6% (86.6–103.5), PsA 97% (85.5–103.2) and RA 96.6% (85.9–102.7)) than using %MRA (median(IQR) for AS 84.0% (54.73–94.74), PsA 85.0% (50.76–98.25) and RA 82.4% (47.24–95.77)). The same pattern was also observed when comparing these two methods by the first biologic received (figure 5A,B). Regardless of which of the two measurements were used, the few patients with PsA (n=7) taking ustekinumab had higher adherence than was seen with any other biologic. Similar adherence results were obtained in the subset with incident users (online supplementary figure 6).

## DISCUSSION

This study is the first real-world cohort using data about SC biologic deliveries, combined with routine data from rheumatology clinics, community pharmacy dispensing,

**Table 1** Baseline characteristics of patients with rheumatic conditions receiving subcutaneous biologics from January 2012 to May 2015 (n=751)

| Rheumatic disease | All | Ankylosing spondylitis | Psoriatic arthritis | Rheumatoid arthritis | P value |
|---|---|---|---|---|---|
| n | 751 | 105 | 227 | 419 | |
| Female, n (%) | 497 (66.2) | 35 (33.3) | 130 (56.5) | 332 (78.7) | $2.2\times10^{-16}$ |
| Age* | | | | | |
| *Age (median, IQR)\** | 53.0 (42.7–60.9) | 47.2 (35.4–55.0) | 48.0 (40.2–57.3) | 56.0 (47.4–64.2) | $4.3\times10^{-14}$ |
| Age by category | | | | | $5.3\times10^{-12}$ |
| 18–34 (%) | 93 (12.4) | 24 (22.9) | 36 (15.9) | 33 (7.9) | |
| 35–49 (%) | 222 (29.6) | 36 (34.3) | 91 (40.1) | 95 (22.7) | |
| 50–64 (%) | 313 (41.7) | 36 (34.3) | 82 (36.1) | 195 (46.5) | |
| 65+ (%) | 123 (16.4) | 9 (8.6) | 18 (7.9) | 96 (22.9) | |
| SIMD† | | | | | 0.326 |
| 1 most deprived (%) | 305 (40.6) | 42 (42.9) | 93 (42.1) | 170 (41.5 | |
| 2 (%) | 132 (17.6) | 21 (21.4) | 44 (19.9) | 67 (16.3) | |
| 3 (%) | 81 (10.8) | 13 (13.3) | 23 (10.4) | 45 (11) | |
| 4 (%) | 93 (12.4) | 15 (15.3) | 25 (11.3) | 53 (12.9) | |
| 5 least deprived (%) | 118 (15.7) | 7 (7.1) | 36 (16.3) | 75 (18.3) | |
| Number of deliveries of biologics | | | | | |
| *Number of deliveries of biologics (median, IQR)* | 6 (3-12) | 7 (3-11) | 6 (3-12) | 6 (3-12) | 0.579 |
| Number of deliveries of biologics by category (%) | | | | | 0.752 |
| 1 | 80 (10.7) | 14 (13.3) | 19 (8.4) | 47 (11.2) | |
| 2–10 | 456 (60.7) | 59 (56.2) | 141 (62.1) | 256 (61.1) | |
| 11–20 | 156 (20.8) | 25 (23.8) | 49 (21.6) | 82 (19.6) | |
| 21+ | 59 (7.8) | 7 (6.7) | 18 (7.9) | 34 (8.1) | |
| Number of different biologics delivered (%) | | | | | 0.108 |
| 1 | 625 (83.2) | 89 (84.8) | 179 (78.9) | 357 (85.2) | |
| 2, 3 or 4 (switchers) | 126 (16.8) | 16 (15.2) | 48 (21.1) | 62 (14.8) | |
| First biologic delivered (%) | | | | | $8.3\times10^{-13}$ |
| Adalimumab | 305 (40.6) | 55 (52.4) | 112 (49.3) | 138 (32.9) | |
| Certolizumab Pegol | 77 (10.3) | 6 (5.7) | <5 | 69 (16.5) | |
| Etanercept | 240 (32) | 24 (22.9) | 82 (36.1) | 134 (32) | |
| Golimumab | 93 (12.4) | 20 (19) | 24 (10.6) | 49 (11.7) | |
| Abatacept | 8 (1.1) | NA | NA | 8 (1.9) | NA‡ |
| Tocilizumab | 21 (2.8) | NA | NA | 21 (5) | |
| Ustekinumab | 7 (0.9) | NA | 7 (3.1) | NA | |
| Follow-up (days), median (IQR) | 529 (284.0–852.0) | 494 (285–838) | 544 (280–831) | 529 (283.5–866.5) | 0.894 |
| Concomitant medication use within one year before index date (%) | | | | | $6.8\times10^{-3}$ |
| Antibacterial use | 48 (6.4) | 6 (5.7) | 11 (4.8) | 31 (7.4) | |
| Oral methotrexate use | 89 (11.9) | <5 | 23 (10.1) | 63 (15) | |
| Any DMARD | 115 (15.3) | <5 | 28 (12.3) | 84 (20) | |
| NSAID | 114 (15.2) | 17 (16.2) | 28 (12.3) | 69 (16.5) | |
| Gluco or corticosteroids | 19 (2.5) | <5 | <5 | 16 (3.8) | |
| Antifungal use | <10 | NA | <5 | 6 (1.4) | NA‡ |

Continued

| Table 1 | Continued | | | | |
|---|---|---|---|---|---|
| **Rheumatic disease** | **All** | **Ankylosing spondylitis** | **Psoriatic arthritis** | **Rheumatoid arthritis** | **P value** |
| Antiviral use | <5 | NA | NA | <5 | |
| Subcutaneous methotrexate use | NA | NA | NA | NA | |
| Concomitant medication use during study period (%) | | | | | $1.0 \times 10^{-7}$ |
| Antibacterial use | 450 (59.9) | 52 (49.5) | 136 (59.9) | 262 (62.5) | |
| Antifungal use | 72 (9.6) | 9 (8.6) | 26 (11.5) | 37 (8.8) | |
| Antiviral use | 53 (7.1) | <5 | 17 (7.5) | 34 (8.1) | |
| Oral methotrexate use | 332 (44.2) | 10 (9.5) | 92 (40.5) | 230 (54.9) | |
| Any DMARD | 493 (65.6) | 12 (11.4) | 139 (61.2) | 342 (81.6) | |
| NSAID | 491 (65.4) | 75 (71.4) | 151 (66.5) | 265 (63.2) | |
| Gluco or corticosteroids | 148 (19.7) | 10 (9.5) | 29 (12.8) | 109 (26) | |
| Subcutaneous methotrexate use | 14 (1.9) | NA | <5 | 11 (2.6) | NA‡ |
| Comorbidities (%)§ | | | | | |
| *Charlson score* | | | | | $<2.2 \times 10^{-16}$ |
| 0 | 163 (21.7) | 38 (36.2) | 75 (33) | 50 (11.9) | |
| 1 | 190 (25.3) | – | – | 167 (39.9) | |
| 2+ | 34 (4.5) | – | – | 28 (6.7) | |
| Unknown | 364 (48.5) | 58 (55.2) | 132 (58.1) | 174 (41.5) | |

\*Age at first delivery,
†From 751 patients, 22 did not have Scottish Index of Multiple Deprivation (SIMD) values. Follow-up is from first delivery to study end date.
‡Comparison was not calculable because numbers in some cells were not available, were zero and/or too small.
§Following the statistical disclosure protocol of the Greater Glasgow and Clyde Safe Haven to avoid attribute disclosure, the cell suppression (primary and secondary) method was used for values in the comorbidities section as some were too small.
DMARD, disease-modifying antirheumatic drugs; NSAID, non-steroidal anti-inflammatory drug.

hospital discharges and mortality records to assess the utilisation of biologics in patients with RA, AS and PsA over a period of 3 years. This study found for all three conditions, approximately 50% of the patients were persistent on their index biologic and that overall adherence during the period of treatment was over 80%. Almost

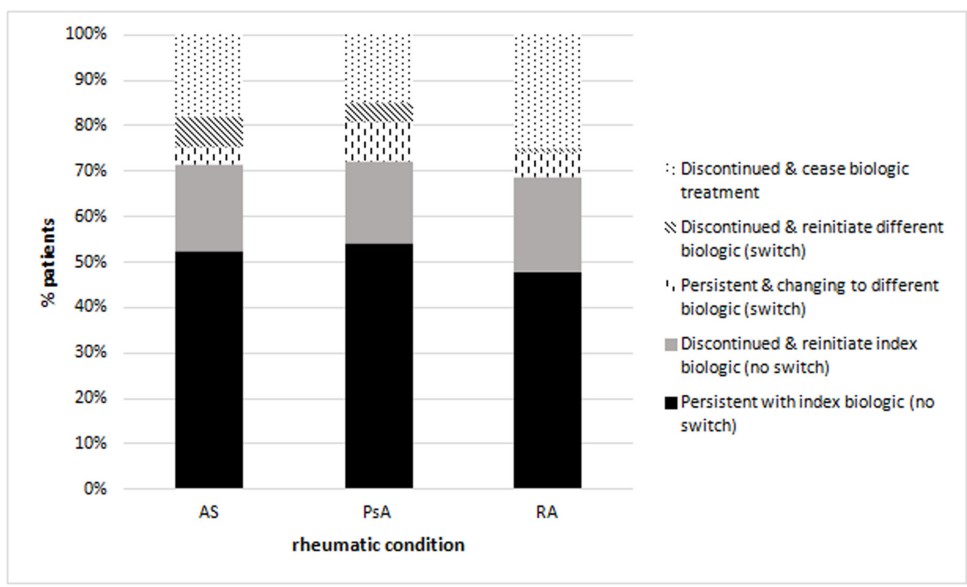

**Figure 2** Persistence and discontinuation (reinitiating or ceasing) of treatment with biologics according to rheumatic condition (n=751).

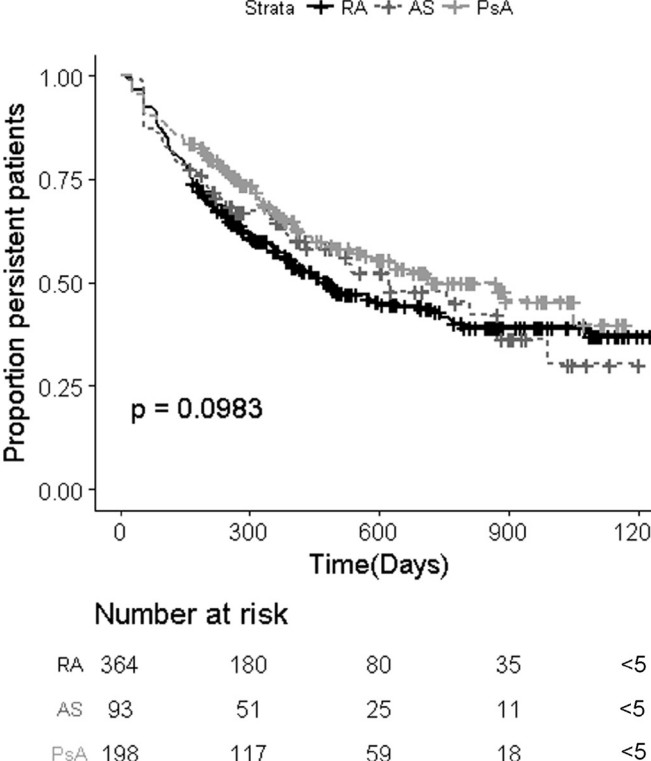

**Figure 3** Crude survival curves comparing persistent patients according to rheumatic condition (n=655). AS, ankylosing spondylitis; PsA, psoriatic arthritis; RA, rheumatoid arthritis.

half of patients who discontinued their index biologic for a period greater than 56 days ultimately reinitiate the same therapy with only a minority switching to another biologic. This highlights the importance of longitudinal data as many of these patients would have been incorrectly counted as cessations in other datasets.

A diverse range of studies have reported on the use of biologics over the last 20 years with the majority focused solely on RA,[19 29 43] and few studies comparing utilisation across the three main inflammatory rheumatological conditions within a single healthcare system. Patient demographic characteristics in our study are similar to those reported in other epidemiological studies in the UK, Denmark, Norway and Germany,[11–16] despite the different methodologies used to create these cohorts.

Linked prescription and administrative health datasets have been used to study the utilisation of biologics in real-world settings in other studies in Europe, USA, Canada and Brazil.[24 28 29 44 45] However, to the best of our knowledge, this is the first study to use routine data generated through 'direct to home' deliveries of biologics as a source of information to assess how patients use these medicines. Most other routine administrative healthcare studies use prescriptions or claims/reimbursement data. In contrast, this homecare delivery service requires active participation of the patient in both ordering and then signing for receipt of the biologic, thereby confirming the biologic was prescribed, the prescription was processed and the patient received the biologic, providing better insight into how patients are using their medicines.

To understand the use of medicines in our cohort, we adopted the taxonomy proposed by ESPACOMP.[27] In summary, the framework enables us to quantify a patient's medicine-taking behaviour in three respects: intensity, continuity and duration—how much of a drug does a patient take, how frequently, and for how long? *Persistence* and *discontinuation* allow us to study continuity and duration of treatment, while *adherence* enables us to study intensity of treatment in the context of the prescribed treatment regimen.

In those studies that examined persistence and discontinuation with biologics across rheumatic conditions, using a consistent in-study method over a comparable time period to our study, persistence was reported as being lower in RA than AS.[45 46] This is similar to our findings where patients with RA were most likely to discontinue their index subcutaneous biologic, potentially reflecting the treat-to-target approach and greater therapeutic options available for RA at the time of this study. Our study also enabled quantification and categorisation of biologic switching, we believe for the first time, in both patients who were persistent regardless of changes in therapy and those who discontinued index therapy and then commenced a new biologic (figure 1).

In our Cox regression analyses, female sex was associated with lower persistence being consistent with several studies.[43 45 46] However, our findings regarding socioeconomic status contradict those of the Machado *et al* study as we have shown higher persistence in patients with lower deprivation levels. There are also discrepancies in the association with the use of methotrexate, as Heiberg *et al* reported a strong association of its concomitant use with persistence in the three rheumatic diseases and we did not find an association. Randomised controlled trials of biologics in patients with PsA have failed to demonstrate superiority of combination therapies.[47 48] Thus, the role of methotrexate as a predictor for persistence remains unclear.

Persistence after the first, second or subsequent courses of biologics is still controversial. A French study found better persistence with first biologic compared with second or third,[49] with similar tendencies reported in Norway by Heiberg *et al*. In contrast, in a large UK national cohort of patients with RA, those who switched to a second biologic had higher persistence,[50] and the study using the Rheumatoid Arthritis Disease-Modifying Anti-Rheumatic Drug Intervention and Utilization Study (RADIUS) registry found similar persistence between first and second biologic.[51] In our study instead of comparing each course, we compared those persisting with their index biologic (first biologic) with those persisting after switching. Our findings were similar to the UK cohort as persistence was higher in those who had switched.

Various studies have examined adherence in a single rheumatoid condition, often RA,[18–24 26–29] but we could only find a single Italian study measuring adherence across RA, AS and PsA,[44] though applying a different method to our study. The challenge in comparing results

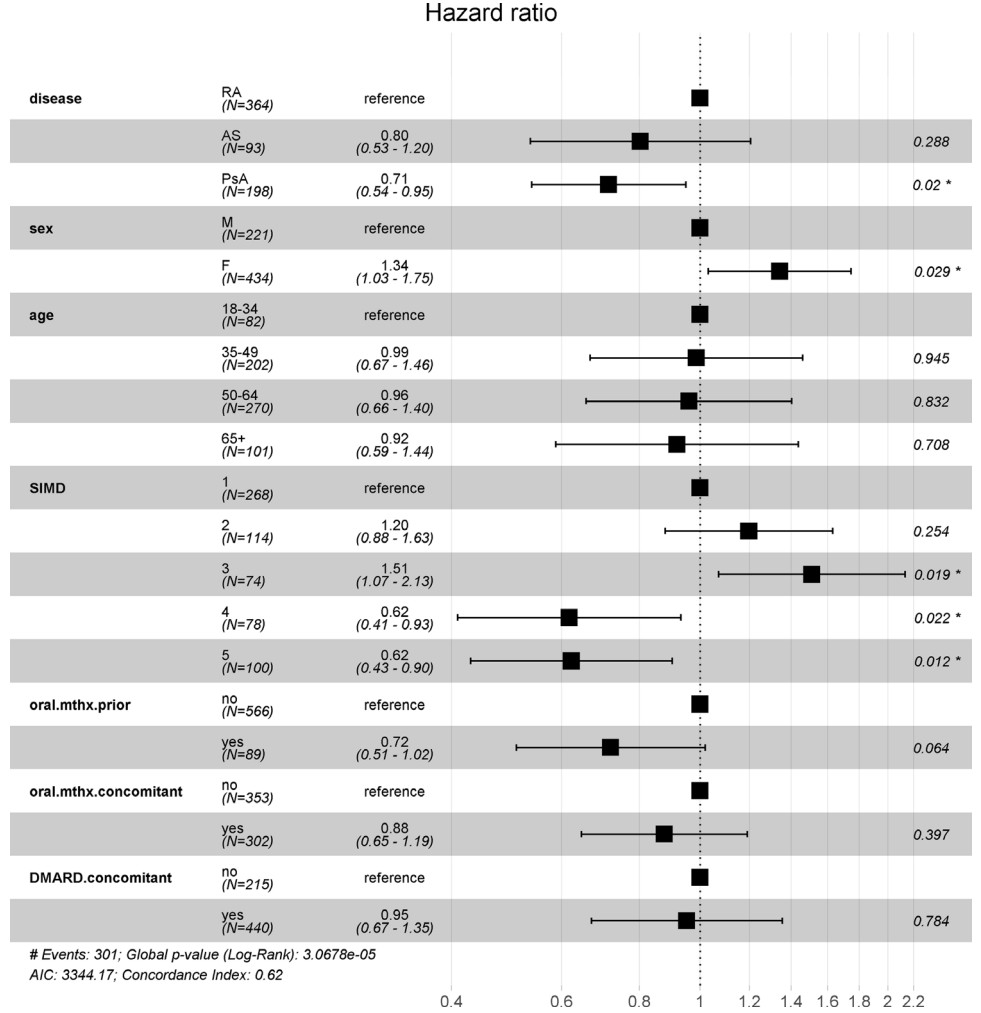

Hazard ratio

**Figure 4** Adjusted HRs for persistence with biologics using Cox regression analysis excluding patients without SIMD values (n=634). SIMD, Scottish Index of Multiple Deprivation quintile.

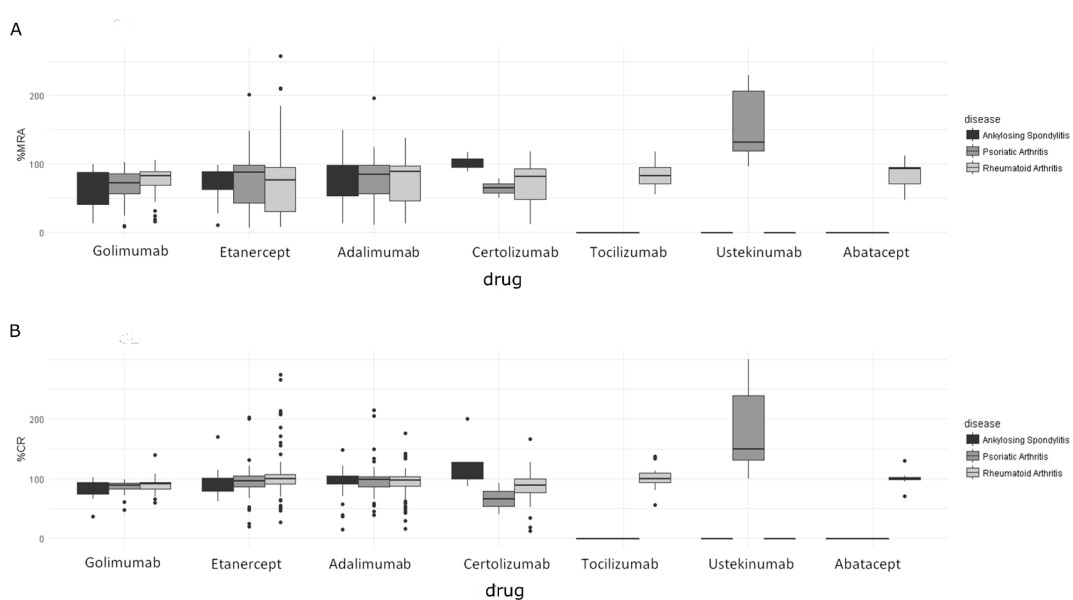

**Figure 5** Adherence to biologics by rheumatic condition according to %MRA (A) and %CR (B). CR, compliance rate; MRA, medication refill adherence.

between studies is often the absence of any detail on how adherence was measured and where specified, multiple measures being used. This has been highlighted in a recent systematic review and meta-analysis by Scheiman-Elazary *et al*[18] of antiarthritis medication in RA, which reported adherence ranging from 10.5% to 98.5% with the meta-analysis generating an adherence rate of 66% (95% CI 58% to 75%), noting that variability may result from different methods and readers should appraise studies according to the validity of the method, the scales and cut-off points. From our experience of using both MRA and CR for adherence measurement, we would recommend the use of CR as it is simple to calculate and includes the period between deliveries, thereby removing any ambiguity arising following the last supply of medicine.

In this study, there were cases where %MRA or %CR was above 100% suggesting some patients could have received more biologics than those expected for that period of time or adjustments with deliveries could be made later on, which were not captured in the study period. The higher adherence to ustekinumab compared with the other biologics in this study may relate in part to the small numbers, a more recent introduction requiring the delivery of two 45 mg vials to achieve the 90 mg dose and a shorter use of this agent at the time of the study, although this observation has also been reported in psoriasis cohorts.[52 53]

This current study has several strengths. This is the first real-world cohort containing type, frequency and dose of subcutaneous biologics delivered directly to patients for these three rheumatic diseases. The cohort is sustainable as it is generated as part of routine clinical care, avoiding the need for additional data collection burden required in a traditional registry model, it can be easily updated and linked to other administrative datasets, has better coverage than individually held clinical data and allows the use of biologics to be studied over time. The sustainability is a key factor, enabling biosimilar and new future innovator biologics to be easily captured as these reach the clinic. This model can also be easily extended to other specialities, including dermatology and gastroenterology, where similar biologic therapies are used. Recording the confirmed delivery of drug directly to patients may reduce some of the uncertainty and limitations associated with using prescription or reimbursement administrative datasets where it cannot be determined if patients actually received the drug. Recall bias was minimised by linking routine data and not based on self-reported persistence or adherence, while selection bias was addressed by including all the patients receiving biologics in the largest Scottish Health Board.

This study also has some limitations. The cohort covers a specific, albeit highly populated, region in Scotland and the generalisability of the results has to be tested. It is a mixed cohort of naive and existing biologic users, which could influence the assessment of medication utilisation. However, the results were similar in the overall

and incident cohorts. By using CR as an adherence measurement, patients discontinuing prior to study completion are not considered. This was overcome in our study by reporting together adherence, persistence and discontinuation. The reasons for discontinuation or treatment interruption are not currently captured in HSD, so it is of limited utility in terms of understanding the factors that influence the use of biologics. History of biologics or subcutaneous methotrexate being used prior to 2012 was not available. However, during the study period, the use of subcutaneous methotrexate was captured within the HSD, recording a relatively low use. The biologics home delivery service only captures subcutaneous biologics and not intravenous agents, such as infliximab or rituximab, so will have missed where patients were switched to these agents. However, the latter are far smaller in volume and may be captured from rheumatology clinic documentation and hospital administrative data where attendance for infusion guarantees receipt of drug.

## CONCLUSION

The long-term evaluation of biologics for the treatment of rheumatic and related diseases is a desirable goal on a clinical level, but it is also a requirement from a pharmacological and economic point of view. Traditional drug or disease registers are expensive, resource intensive and difficult to sustain, so strategies using linked administrative and routine clinical data provide the best opportunity to capture the true picture of biologic use. Additionally, a homecare delivery database may reduce some of the uncertainties associated with other administrative databases.

In moving forward, we strongly advocate that future studies should consider application of the ESPACOMP framework[25] to better understand the use of biologics, that is, the combination of discontinuation, persistence and adherence metrics, thus leading to a more consistent approach that enables more comparability between studies.

**Author affiliations**
[1]Health Data Research Scotland, Strathclyde Institute of Pharmacy and Biomedical Sciences, University of Strathclyde, Glasgow, UK
[2]School of Public Health, Imperial College London, London, United Kingdom
[3]Department of Mathematics and Statistics, University of Strathclyde, Glasgow, UK
[4]Institute of Infection Immunity and Inflammation, University of Glasgow, Glasgow, UK
[5]Strathclyde Institute of Pharmacy and Biomedical Sciences, University of Strathclyde, Glasgow, UK
[6]Medicines Information, Pharmacy Department, Glasgow Royal Infirmary, Glasgow, UK
[7]Division of Clinical Pharmacology, Karolinska Institutet, Karolinska University Hospital, Stockholm, Sweden
[8]Division of Public Health Pharmacy and Management, School of Pharmacy, Sefako Makgatho Health Sciences University, Ga-Rankuwa, Pretoria, South Africa
[9]Social Pharmacy, School of Pharmacy, Universidade Federal de Minas Gerais, Belo Horizonte, Brazil
[10]Information Services Division, NHS National Services Scotland, Edinburgh, UK

# Open access

**Acknowledgements** Farr Institute @ Scotland, GGC Safe Haven team, Norman Lannigan and Kevin Milne for providing the Homecare Database and allow its use for research purposes. FdAA acknowledges the support from Fundação de Amparo à Pesquisa do Estado de Minas Gerais (FAPEMIG), the Minas Gerais State Research Foundation, Brazil.

**Contributors** All authors contributed to study design. SA-M extracted, prepared the data and performed the analyses. SA-M drafted the manuscript, with revisions by KK, SS, YS, BG, AMA, FdAA and MB.

**Funding** We acknowledge the support from The Farr Institute @ Scotland. The Farr Institute @ Scotland is supported by a 10-funder consortium: Arthritis Research UK, the British Heart Foundation, Cancer Research UK, the Economic and Social Research Council, the Engineering and Physical Sciences Research Council, the Medical Research Council, the National Institute of Health Research, the National Institute for Social Care and Health Research (Welsh Assembly Government), the Chief Scientist Office (Scottish Government Health Directorates) and the Wellcome Trust (MRC Grant No: MR/K007017/1).

**Competing interests** SS has received honoraria from Janssen, Novartis, AbbVie, Pfizer, UCB, Celgene and Boehringer-Ingelheim, and research funding from UCB, Celgene, Boehringer-Ingelheim, BMS and Pfizer.

**Patient consent for publication** Not required.

**Ethics approval** Ethics permission was not required as only non-identifiable routine data were used for this study. The study was approved by the National Health Service GGC Safe Haven Local Privacy Advisory Committee (project number: GSH/15/MT/001). No patient identifiers were available to the study team, and all data were accessed via the GGC Safe Haven.

**Provenance and peer review** Not commissioned; externally peer reviewed.

**Data sharing statement** All data supporting this research are openly available to researchers based in recognized UK institutions from http://www.nhsggc.org.uk/about-us/professional-support-sites/nhsggc-safe-haven/your-research/available-datasets. For further information contact the Greater Glasgow and Clyde Safe Haven by emailing safehaven@ggc.scot.nhs.uk.

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
