## [Reviewer comments · BMJ Open]

ARTICLE DETAILS

TITLE (PROVISIONAL)	Discontinuation, persistence and adherence to subcutaneous biologics delivered via a homecare route to Scottish adults with rheumatic diseases: a retrospective study.
AUTHORS	Alvarez-Madrado, Samantha; Kavanagh, Kimberley; Siebert, Stefan; Semple, Yvonne; Godman, Brian; Maciel Almeida, Alessandra; Acurcio, Francisco; Bennie, Marion

VERSION 1 – REVIEW

REVIEWER	Dr Gouri Koduri Southend University Hospital , United Kingdom
REVIEW RETURNED	07-Nov-2018

GENERAL COMMENTS	1. Medication adherence and persistence is a worldwide problem and continues to be challenge in chronic diseases. It is a very good study on drug adherence and discontinuation in several inflammatory conditions using the same analysis.2. Although it is retrospective study both incident and prevalent users were studied although not ideal method. The results might be different in a prospective study.3. Statistical methods are reasonable. For Persistence KM and Cox regression models were used. MRA is also a good method for adherence.4. However reviewing prescription refill records using electronic databases doesn't always provide clues to non-adherence .Electronic Database analysis fails to identify partial adherence and barriers. This study didn't look at personal or psychosocial factors or broader understanding of the determinants of non – adherence or discontinuation across different conditions. It would have been interesting to see the reasons for discontinuation or treatment interruption. It might be worth looking at these factors if the data is available.5. This could be replicated in a prospective longitudinal study combined with health economics.
---

REVIEWER	Gerald Lebovic St. Michael's Hospital, Canada
REVIEW RETURNED	19-Nov-2018

GENERAL COMMENTS	Very interesting and informative paper. This study has the added benefit of having a rich source of the prescription data. I have provided some comments to address. Page 5 Lines 6-10: With regard to exclusions it would seem people who had their first biologic prior to the date of approval are
--

being left truncated (as well as possibly left censored) from the analysis. How was this dealt with? How many subjects was this? With regard to the missing records. Were subjects excluded if they had records missing once or all data was missing? How many records were missing and how was this handled in the analysis? If warranted, this should also be mentioned in the limitations.

Page 5 lines 51-57: Data analysis - A few questions with regard to the modeling approach:

1. Were the proportional hazards assumption checked? If so, how and were they met?
2. It seems that you have more than the 1 outcome persistence vs. not. Currently you have the possibility of a switch from the index biologic as a covariate and it is unclear to me how this was handled in the model. What about using a competing risk model to account for the possibility of switching to another biologic (a competing risk) ? What about death? Were there any in this cohort and if yes, how was this addressed in the model?

Page 6: Lines 10-12: Definition of CR is unclear.

Page 6: Line 14 - cite reference for R software

Page 6: results Lines 29-50. You mention characteristics of the population (e.g. RA were older etc.) however I don't know whether these were statistically meaningful. I couldn't find p-values in the results nor the table. If you want to describe the cohort and mention that there are differences between the groups, some statistical test should be conducted.

Page 7 - Table 1: What was the distribution of deliveries of biologics when treated as a continuous variable? (median and (range or IQR.))

Page 8 (table contd): Follow-up time: How is it that the range for the RA group is wider than the range of all subjects? The max. in the RA group is larger than the max overall...

Page 9 (table contd) Comorbidities. For the AS group there is an entry <5 that can be determined exactly by subtracting the total from the other entries. Needs to be fixed.

Page 10: Lines 7-28: Use absolute numbers and percent for consistency. It is confusing to base some percents on subsets and try and follow the corresponding figure. Results should also go in the same order as the figure for ease of interpretation.

Page 10: Line 34 - Figure 2 seems to show the PH assumption is violated in the earlier part of the study. Log rank test therefore has less power. Also this relates to my earlier comment about the Cox PH model.

Page 11 line 5: Was imputation considered for the missing SIMD values? Was the missing data mechanism assessed?

Page 11 - Adherence section. I am not sure how the %CR was over 100% please explain. Similarly Figures 5a and 5b show subjects above 200% MRA or CR. How is this computed? Please explain.

	Page 13: Lie 51-52. I would think the limitation of of patients' response not being captured is a large one? Do you know if patients were no longer prescribed for whatever reason? Figure 1 (I think...) KM for persistence. Please include a table with the number at risk. Figure 5 (I think): can't read most of the text - too small font on printed page Supplementary file: Figure 1 - Venn Diagram appears incorrect. there should be two circles: One for HSD and one for RAD and then appropriate overlapping for the OR and AND statements. It looks as though everything is in HSD. The OR group looks as though it is a subset of the AND group. Supplementary Table 1 - page 26 Follow-up time. Range for all subjects has a minimum above the RA group and a maximum below the AS group. How is this? Some minor comments: Figures were hard to follow-no clear labels. Legends in KM figures need to be adjusted - look awkward. Page 5 Line 16 missing period. single indexToAssess Page 5 line 55: oral methotrexate a year prior - seems like 1 year before. should this be within a year? Page 13 Line 47: The study has also some limitations - should be "also has"
--	---

VERSION 1 – AUTHOR RESPONSE

Comments: Reviewer 1	Authors' response
1.Medication adherence and persistence is a worldwide problem and continues to be challenge in chronic diseases. It is a very good study on drug adherence and discontinuation in several inflammatory conditions using the same analysis.	Thank you for this comment. We do believe that as the use of biologics keeps increasing in several rheumatic conditions, it is relevant to compare the discontinuation, persistence and adherence over time.
2.Although it is retrospective study both incident and prevalent users were studied although not ideal method. The results might be different in a prospective study.	We agree one of the limitations of our study is to be a retrospective mixed cohort of incident and prevalent users as described in the Discussion section. The results of this study will enable us to design a prospective study in the future.
3.Statistical methods are reasonable. For Persistence KM and Cox regression models were used. MRA is also a good method for adherence.	We agree on the statistical methods used as they allow to compare our results with other studies using administrative health datasets in Europe, USA, Canada and Brazil as mentioned in the Discussion section.

4. However reviewing prescription refill records using electronic databases doesn't always provide clues to non-adherence. Electronic Database analysis fails to identify partial adherence and barriers. This study didn't look at personal or psychosocial factors or broader understanding of the determinants of non-adherence or discontinuation across different conditions. It would have been interesting to see the reasons for discontinuation or treatment interruption. It might be worth looking at these factors if the data is available.	The limitation of not currently capturing the reasons for discontinuation or treatment interruption in HSD has been re-worded in the Discussion section to clarify the data are not available.
---	---

5. This could be replicated in a prospective longitudinal study combined with health economics.	Thank you, this study design is considered in the future studies using HSD.
Comments: Reviewer 2	

Page 5 Lines 6-10: With regard to exclusions it would seem people who had their first biologic prior to the date of approval are being left truncated (as well as possibly left censored) from the analysis. How was this dealt with? How many subjects was this? With regard to the missing records. Were subjects excluded if they had records missing once or all data was missing? How many records were missing and how was this handled in the analysis? If warranted, this should also be mentioned in the limitations.	There was only one patient who had its first biologic prior to the date of approval and it was excluded from the cohort. It was considered that having this patient excluded would not impact the analyses. To clarify this, the number of patients excluded has now been added to Figure 1 and is also mentioned in the third paragraph of the section 'Study Population'. Two subjects were excluded as all their records were missing dose, frequency or how often the biologics were delivered to enable calculations of weeks covered per delivery in the rest the cohort. This is now modified in Figure 1 and the third paragraph of the section 'Study Population'. When patients had missing records in one variable, their records were kept . This is stated in the fourth paragraph of the section 'Study Population' and the legend of Table 1 where it is mentioned 22 patients did not have Scottish Index of Multiple Deprivation (SIMD) values. However, as those with missing SIMD values were a few (3%) they were excluded when using SIMD as a covariate for the adjusted hazard ratios in Figure 4. As the number of patients with missing records was small and did not have a substantial impact in the results of the study it was not considered as a limitation.
Page 5 lines 51-57: Data analysis - A few questions with regard to the modeling approach:  1. Were the proportional hazards assumption checked? If so, how and were they met? 2. It seems that you have more than the 1 outcome persistence vs. not. Currently you have the possibility of a switch from the index biologic as a covariate and it is unclear to me how this was handled in the 	For the modelling approach:  1. The proportional hazards (PH) assumption were checked using Grambsch and Therneau method in R both in the crude and adjusted model stratifying by those who used the index biologic or switched. The PH assumption was not violated in any of the models. Results of the corresponding global p-values were added to the Discontinuation and persistence section, and Figure 4 with the Adjusted Hazard Ratios using the stratifying variable has been updated.

model. What about using a competing risk model to account for the possibility of switching to another biologic (a competing risk) ? What about death? Were there any in this cohort and if yes, how was this addressed in the model?	2. We used the concepts of discontinuation and persistence advocated by Vrijens et al. and endorsed by the European Society for Patient Adherence, Compliance and Persistence (ESPACOMP). Patients were either persistent or discontinued (because there was a considerable gap of more than 56 days between deliveries or no more deliveries were received). Both persistence and discontinuation were defined regardless of whether or not patients switched to another biologic as mentioned in the Data Analyses section. The different subgroups are shown in Figure 2. Competing risks would apply to mutually exclusive events. In this case persistence and switching are not mutually exclusive. For this study it was considered more appropriate to stratify according to those who used the index biologic or switched (Figure 4 with the Adjusted Hazard Ratios using the stratifying variable has been updated). In the section Study population it was described all patients were followed-up until death, migration to a primary care practice outside Scotland or end date of the study (May 2015), whichever occurred first. A sentence has been added in Data analysis section indicating that patients who died or migrated were censored in the survival analyses.
Page 6: Lines 10-12: Definition of CR is unclear.	The definition of CR is included at the end of the Data analyses section. CR describes the exposure between first and last prescription delivered (total days' supply excluding last delivery/days first up to but not including last delivery) x 100. Instead of being an overall adherence percentage like MRA, CR shows the adherence value for a period between deliveries. As stated in the discussion it can remove any ambiguity arising following the last supply of medicine. Its limitation has been included in the Discussion.
Page 6: Line 14 - cite reference for R software	The most recent version of the R software has been used to update the manuscript and its corresponding reference has been included.

Page 6: results Lines 29-50. You mention characteristics of the population (e.g. RA were older etc.) however I don't know	Chi-square and Fisher's exact tests were used for categorical variables, and Kruskal-Wallis for continuous variables (Data analyses section). The corresponding p-
whether these were statistically meaningful. I couldn't find p-values in the results nor the table. If you want to describe the cohort and mention that there are differences between the groups, some statistical test should be conducted.	values for global comparisons between all groups are now included in Table 1 and supplementary Table 1.
Page 7 - Table 1: What was the distribution of deliveries of biologics when treated as a continuous variable? (median and (range or IQR.))	The median and IQR for deliveries of biologics is now included in Table 1 for all the patients and for each rheumatic disease. The values of all are similar.
Page 8 (table contd): Follow-up time: How is it that the range for the RA group is wider than the range of all subjects? The max. in the RA group is larger than the max overall.	The maximum follow-up time for the RA group and all the subjects was 1231 days. However, the reported values for this variable in Table 1 were median and IQR, not maximum.
Page 9 (table contd) Comorbidities. For the AS group there is an entry <5 that can be determined exactly by subtracting the total from the other entries. Needs to be fixed.	Thank you for highlighting this. Following the statistical disclosure protocol of the Greater Glasgow and Clyde Safe Haven to avoid attribute disclosure, absolute numbers with cell values <5 cannot be shown. An alternative method to present the data in Table 1 and Supplementary Table 1 was using cell suppression (primary and secondary) for unsafe cells. This change has been described in the Data Analyses section and the reference from the NHS National Services Scotland Statistical Disclosure Control Protocol has been added.
Page 10: Lines 7-28: Use absolute numbers and percent for consistency. It is confusing to base some percents on subsets and try and follow the corresponding figure. Results should also go in the same order as the figure for ease of interpretation.	Those lines have been modified and now they are based on 100% for each rheumatic disease in the overall or the incident cohort. The results are described in the same order as Figure 2 and supplementary figure 2 to facilitate interpretation. As a result of this change, the percentages in the abstract were also modified. Following the statistical disclosure protocol of the Greater Glasgow and Clyde Safe Haven absolute numbers with cell values <5 cannot be shown.

Page 11 line 5: Was imputation considered for the missing SIMD values? Was the missing data mechanism assessed?	Imputation was not considered for SIMD values and the missing data mechanism was not assessed. Those values were missing for 3% of the subjects and there were kept in the adjusted hazard ratios. We believe it is unlikely this small proportion of subjects would have a considerable impact in the results.
Page 11 - Adherence section. I am not sure	Both, %CR and %MRA are the result of dividing a
how the %CR was over 100% please explain. Similarly Figures 5a and 5b show subjects above 200% MRA or CR. How is this computed? Please explain.	supply (numerator) by time (denominator). When there is a possible oversupply of medicine in that period of time, the % could be more than 100% or even 200%. A couple of sentences were added in the Discussion section to explain possible causes of receiving more biologics than those expected.
Page 13: Lie 51-52. I would think the limitation of of patients' response not being captured is a large one? Do you know if patients were no longer prescribed for whatever reason	Unfortunately the HSD does not currently capture the reason for stopping the prescription of any biologic.
Figure 1 (I think...) KM for persistence. Please include a table with the number at risk.	An updated figure 3 of KM for persistence with at risk table is included.
Figure 5 (I think): can't read most of the text - too small font on printed page	The resolution of Figure 5 has been updated so text can be read.
Supplementary file: Figure 1 - Venn Diagram appears incorrect. there should be two circles: One for HSD and one for RAD and then appropriate overlapping for the OR and AND statements. It looks as though everything is in HSD. The OR group looks as though it is a subset of the AND group.	Originally the number of patients only in RAD was not included as they were not part of the validation study. For clarity, this number has now been added and Figure 1 in supplementary file modified.
Supplementary Table 1 - page 26 Followup time. Range for all subjects has a minimum above the RA group and a maximum below the AS group. How is this?	The maximum follow-up time for the RA group and all the subjects was 1062 days. However, the reported values for this variable in Supplementary Table 1 were median and IQR, not maximum.
Minor comments	
Figures were hard to follow-no clear labels.	The resolution and labels from figures has been updated.
Legends in KM figures need to be adjusted	Legends have been updated in figure 3 of KM for persistence.

Page 5 Line 16 missing period. single indexToAssess	Missing period has been added to this line.
Page 5 line 55: oral methotrexate a year prior - seems like 1 year before. should this be within a year?	This now reads ' use of oral methotrexate in the 12 months before to the index date'. For clarity the wording in Table 1 and Supplementary Table 1 has also been changed to 'Concomitant medication use within one year before index date'.
Page 13 Line 47: The study has also some limitations - should be "also has"	This now reads ' This study also has some limitations'.

VERSION 2 – REVIEW

REVIEWER	Gerald Lebovic St. Michael's Hospital
REVIEW RETURNED	10-Apr-2019

GENERAL COMMENTS	The authors have addressed my concerns and the paper is well written. I have 2 minor clarifications: n the response to Reviewer 2 1) Page 6: lines 10-12. the definition of CR is written as "total days' supply excluding last delivery/days first up to but not including last delivery) x 100. What does the "first" mean? Is it days up to but not including last delivery? What is the word first adding 2) In the response to Reviewer 2 Page 8 (table contd), the authors responded that Table 1 is median and IQR. In my version Table 1 has listed median and range. Only Table 1 in the supp. section has median and IQR. Please fix this.
--

VERSION 2 – AUTHOR RESPONSE

Comments: Reviewer 2	Authors' response
1) Page 6: lines 10-12. the definition of CR is written as "total days' supply excluding last delivery/days first up to but not including last delivery) x 100. What does the "first" mean? Is it days up to but not including last delivery? What is the word first adding	Thank you for the comments. To clarify the meaning of "first" in the definition of CR in page 6 the sentence has been modified to: "In contrast, CR describes the exposure between first and last prescription delivered (total days' supply excluding last delivery/days from first delivery up to but not including last delivery) x 100."
2) In the response to Reviewer 2 Page 8 (table contd), the authors responded that Table 1 is median and IQR. In my version Table 1 has listed median and range. Only	Table 1 (now in page 8-11) has been modified to have median and IQR as in the supp.section, mainly the variable Follow-up (days), median (IQR).

Table 1 in the supp. section has median and IQR. Please fix this.	
3. The ScholarOne system does not allow to amend the details of two co-authors	The second co-author currently reads "Kimberly Kavanagh" please change to "Kimberley Kavanagh". For the third co-author "Stefan Siebert" currently the Institution cannot be modified (Glasgow, Glasgow City, UK) please change to" Institute of Infection Immunity and Inflammation, University of Glasgow,UK".